# Determinants of Different Aspects of Upper-Limb Activity after Stroke

**DOI:** 10.3390/s22062273

**Published:** 2022-03-15

**Authors:** Bea Essers, Camilla Biering Lundquist, Geert Verheyden, Iris Charlotte Brunner

**Affiliations:** 1Department of Rehabilitation Sciences, KU Leuven, 3001 Leuven, Belgium; geert.verheyden@kuleuven.be; 2Hammel Neurorehabilitation Centre and University Research Clinic, 8450 Hammel, Denmark; iris.brunner@rm.dk; 3Department of Clinical Medicine, Aarhus University, 8000 Aarhus, Denmark

**Keywords:** stroke, upper-limb sensor activity, determinants

## Abstract

We examined factors associated with different aspects of upper-limb (UL) activity in chronic stroke to better understand and improve UL activity in daily life. Three different aspects of UL activity were represented by four sensor measures: (1) contribution to activity according to activity ratio and magnitude ratio, (2) intensity of activity according to bilateral magnitude, and (3) variability of activity according to variation ratio. We combined data from a Belgian and Danish patient cohort (*n* = 126) and developed four models to determine associated factors for each sensor measure. Results from standard multiple regression show that motor impairment (Fugl–Meyer assessment) accounted for the largest part of the explained variance in all sensor measures (18–61%), with less motor impairment resulting in higher UL activity values (*p* < 0.001). Higher activity ratio, magnitude ratio, and variation ratio were further explained by having the dominant hand affected (*p* < 0.007). Bilateral magnitude had the lowest explained variance (adjusted *R*^2^ = 0.376), and higher values were further associated with being young and female. As motor impairment and biological aspects accounted for only one- to two-thirds of the variance in UL activity, rehabilitation including behavioral strategies might be important to increase the different aspects of UL activity.

## 1. Introduction

A main goal of upper-limb rehabilitation after stroke is being able to perform everyday tasks [1]. Most everyday tasks consist of bilateral actions, using both upper limbs (ULs) [2]. In order to facilitate daily tasks, it will, thus, be important to engage both the affected and the unaffected UL in daily life activities.

A reliable and valid way of measuring the UL activity in daily life is using wrist-worn accelerometry [3,4]. Previous accelerometer studies in healthy older adults showed that most activities are performed bimanually, with both ULs active to a similar degree [5,6]. UL movements in the chronic phase post stroke appear quite different; people not only use the affected UL less than the unaffected UL, but also at a lower intensity and with less variation [6,7,8]. Different aspects of UL activity are captured by different sensor measures. To develop strategies to increase UL activity in the chronic phase post stroke, it will be important to understand the different sensor measures and which aspects of UL activity they reflect [4].

First, there are the so-called sensor measures of symmetry [9]. These are ratios of acceleration characteristics between the affected and unaffected UL capturing how one UL moves compared to the other. The simplest and most studied variable of UL symmetry is the activity ratio [10]. It compares the UL activity duration of the affected versus the unaffected UL for a given time. The activity ratio is reliable and valid, and norm values for different age groups have been established [5,11]. These norm values allow analyzing the divergence from normal use in populations with stroke [9,12]. Although the activity ratio informs about the relative contribution from each UL to the activity, this information is incomplete. For example, if the affected UL is active for 2 h and the unaffected UL is active for 4 h during the wearing period, the activity ratio would be 2/4 = 0.5. However, this value could be obtained if both ULs were active alone (unilateral movements) or simultaneously (bilateral movements) [13]. Thus, the activity ratio does not accurately reflect the actual bilateral UL activity. Therefore, a newer method to express bilateral activity was developed [14]. In this method, the magnitude ratio was calculated, which quantifies the contribution from both ULs for each second of UL activity.

The contribution of both ULs to an activity is not the only variable of interest when looking at daily task performance. Not only are some tasks performed unilaterally and others more bilaterally, but different tasks are also performed at a different intensity. Therefore, another important aspect of UL activity is the intensity, quantified by the bilateral magnitude [14].

Lastly, next to the contribution to and intensity of UL activity, there is the variability of UL movement, quantified by the variation ratio. Although this variable has been explored to a lower extent [4], it may be important in assessing daily life UL activity as it is correlated with movement quality [15]. It seems that individuals who move the UL more in daily life in terms of time and variability tend to move with fewer compensations [15].

Several studies investigated factors influencing daily UL activity post stroke [16,17,18,19,20,21]. Generally, it was found that patients with higher UL motor impairment had a lower (contribution to the) activity compared to patients with a lower motor impairment [6,17,18]. However, improvement of affected UL motor impairment in persons at 6 months and 1 year after stroke did not necessarily translate into a significant increase in UL activity [17,19]. This indicates that other factors might influence UL activity post stroke. UL activity was, for instance, decreased in a group of persons in the chronic phase post stroke with the nondominant hand affected [16]. In contrast, whether the dominant or nondominant hand was affected did not seem to influence UL intensity [16]. Furthermore, gender might also play a role, as being female was associated with less UL activity after rehabilitation [20].

Most studies focused mainly on activity ratio, in single associative analyses [16]. Furthermore, studies were performed in the acute phase post stroke [21] or using small samples. Lastly, to our knowledge, no study investigated the factors influencing variation ratio. Therefore, the purpose of this exploratory study was to establish associative factors of different aspects of UL activity in a larger group of adults with chronic stroke. We hypothesized that UL impairment would be a significant determinant of all aspects of daily UL activity and that other factors would contribute, such as gender and whether the dominant hand is affected. Improved understanding of what influences different aspects of UL activity in the chronic phase post stroke might further guide strategies to improve UL activity and, thus, enhance the performance of everyday tasks in the chronic phase post stroke.

## 2. Materials and Methods

### 2.1. Study Design and Participants

This was an exploratory study of cross-sectorial data obtained from two cohort studies, performed in two different countries. We combined data collected at either 6 months post stroke in Denmark [12] or later than 6 months post stroke in Belgium [18]. The inclusion criteria in both studies were persons with first or recurrent stroke, more than 18 years old, and with cognitive ability to comply with the examinations. The exclusion criteria were a musculoskeletal and/or other neurological disorder such as head injury or Parkinson’s disease that influenced UL function. From the datasets, common characteristics were collected, as well as the observed UL motor impairment as assessed with the upper extremity subscale of the Fugl–Meyer Assessment (FMA-UE) [22]. Upper-limb activity data were collected with identical wrist-worn accelerometers.

### 2.2. Procedure

Demographic and health information was provided from medical records in Denmark or self-report during home visits in Belgium. This information includes age at the time of inclusion, gender, time post stroke, lateralization of symptoms, and pre-stroke hand dominance.

Then, the observed UL motor impairment was assessed with the FMA-UE [22]. The FMA-UE consists of 33 items, with a total score between 0 and 66 and higher scores indicating lower motor impairment. The FMA-UE is internationally recommended [23] and the psychometrics of the scale are well established [24,25,26]. Outcome assessors were experienced therapists instructed in the scale, and, to further ensure reliability, a scoring manual was used [27].

We used wrist-worn accelerometry to measure UL activity as the reliability and validity are well established in adults with stroke [11,13,14]. Participants were instructed on how to don the accelerometers and helped by the research therapists or relatives to do so. The accelerometers were worn while the participants performed their daily routines.

### 2.3. Accelerometry

The accelerometers (ActiGraph GT3x+ and wGT3-BT Activity Monitors) were worn from 8:00 a.m. to 8:00 p.m. on a weekday in Denmark and from 12:00 p.m. until 12:00 p.m. 72 h later in Belgium. Participants were instructed to go about their normal, daily routines without changing behavior or trying to increase their UL activity. After the wearing period, accelerometers were returned to the research lab or picked up by the researcher. Accelerations were recorded along three axes at a predefined frequency (30 and 50 Hz) and converted into activity counts (0.001664 g/count) [10,28]. Activity counts across the three axes were combined into a single value, called a vector magnitude (VM) $\sqrt{x^{2}+y^{2}+z^{2}}$.

ActiLife 6 software (Actigraph Inc., Pensacola, FL, USA) was used to visually inspect the data and ensure that accelerometers were worn for the planned time and functioned properly. To have comparable intervals for the two cohorts, we then isolated 8:00 a.m. to 8:00 p.m. intervals for both cohorts in Matlab R2020a (Mathworks, Nattick, IL, USA) and exported them to IBM SPSS Statistics for Windows, version 27.0 (IBM Corp., Armonk, NY, USA) or Stata Statistical Software: Release 16 (StataCorp. 2019. College Station, TX, USA: StataCorp LLC.). Afterward, we calculated different UL sensor measures as described by Lang et al. [29] and in the work by Urbin et al. [7,13]. The definition of each sensor measure is described in Table 1.

**Table 1 sensors-22-02273-t001:** Definitions for sensor measures.

Activity ratio [3,29,30]	Ratio of the total activity hours of the affected UL compared to the unaffected UL, reflecting the relative contribution of the affected UL over the entire monitoring period.
Magnitude ratio [6,13,14]	Contribution of each arm to activity, calculated for each second of activity.
Bilateral magnitude [6,14]	Summed intensity of activity across both arms, calculated for each second of activity.
Variation ratio [7,13]	Ratio of acceleration variability of the affected UL compared to the unaffected UL, reflecting the relative variability in the affected UL over the entire monitoring period.

In summary, we calculated (1) hours of affected and unaffected UL activity by summing all seconds when the VM was nonzero and converting them to hours, (2) the activity ratio by dividing total hours of affected UL activity by total hours of unaffected UL activity, (3) the magnitude ratio by taking the natural log of the VM of the affected UL divided by the VM of the unaffected UL, whereby values greater than 7 and less than −7 were replaced by 7 and −7, (4) the bilateral magnitude by summing the VM from both limbs, and (5) the variation ratio by dividing the standard deviation of the VM of the affected UL by the standard deviation of the VM of the unaffected UL.

The activity ratio and the magnitude ratio quantify the contribution to the activity of one limb versus the other, whereas the bilateral magnitude quantifies the intensity of activity across both ULs. The activity ratio gives a general overview of the contribution to the activity over the entire wearing period, in which an activity ratio of 0.5 indicates that the affected UL was active 50% of the time the unaffected UL was active. The magnitude ratio and bilateral magnitude on the other hand give insight into the UL activity for every second of data. For the magnitude ratio, values go from −7 (unilateral unaffected UL movement) to 7 (unilateral affected UL movement), in which every value between −7 and 7 indicates bilateral UL movement. Negative values indicate the unaffected UL contributed more to the activity, positive values indicate the affected UL contributed more, and a value of 0 indicates that both ULs contributed to the activity to the same extent. For the bilateral magnitude, values can go from small values for low intense unilateral activities (e.g., 6 for writing) to high values for high-intensity bilateral activities (e.g., 427 for putting boxes on a shelf at shoulder height) [14]. Lastly, the variation ratio quantifies the variability of the activity of one limb versus the other, in which values near 1 indicate that the acceleration variability is equally spread in the affected UL compared to the unaffected UL. Values <1 indicate a greater spread of affected acceleration relative to unaffected accelerations, and the opposite is true for values >1 [13].

### 2.4. Statistical Analysis

Normality was checked for all variables with Shapiro–Wilk and histograms. Normally distributed variables were summarized by the mean and standard deviation (SD), while not normally distributed variables were summarized by medians with first quartile (Q1) and third quartile (Q3), and frequencies with percentages for counts. Data were analyzed with SPSS Version 27.0 with the level of statistical significance set two-tailed at *p* < 0.05. This study conformed to the STROBE guidelines and reported the required information accordingly.

### 2.5. Regression Models

We developed models to determine variables associated with different aspects of daily UL use, i.e., contribution of the ULs to activity, intensity, and variation of UL activity. Measures of contribution were the activity ratio and magnitude ratio, the measure of intensity was the bilateral magnitude, and the measure of variation was the variation ratio. Activity ratio, magnitude ratio, bilateral magnitude, and variation ratio were analyzed as the dependent variables. Independent variables assessed for their contribution to the models were FMA-UE, dominant side affected, lesion side, lesion type, age, gender, time post stroke, and country. These potential variables were chosen on the basis of the results of previous studies [16,17,20], clinical reasoning, and availability.

To maintain adequate power for the statistical analysis, we complied with the events per variable rule, which calls for a sample size of at least *n* = 114 (50 plus eight times the number of independent variables, i.e., 50 + 8 × 8) [31,32]. Furthermore, all necessary assumptions for generalized linear models, including linearity, equality of variance, and normally distributed residuals were visually inspected for all models. The presence of multicollinearity was examined by the tolerance and variance inflation factor (VIF) for each independent variable. A tolerance value of less than 0.10 or a VIF value of above 10 indicates that the multiple correlation with other variables is high, suggesting the possibility of multicollinearity, which was, therefore, not accepted [31]. Lastly, we checked the Mahalanobis distance (*p* < 0.001 criterion) and Cook’s distance for possible problems with outliers and influential data points, in which points with a Cook’s distance <1 were considered problematic [31].

As this was an exploratory study, we performed standard multiple regression in which, for each dependent variable, all potential determining variables were simultaneously entered into the equation. Next, we repeated the regression with only those determining variables that were significant determinants in the model with all variables included (*p* < 0.05). However, it might be that, in this model, which included all variables, beta weights are not statistically significant due to multicollinearity among determinants in the same model. Therefore, in the case of multicollinearity, different models were made for the dependent variable, whereby each model included one of these correlated independent variables. Lastly, for each dependent variable, the model with the lowest number of significant determining variables and the largest explained variance for the dependent variable was kept.

The strength of the association of the dependent and independent variables was assessed by the size of the adjusted *R*^2^ and the standard error of the estimate. The contribution of each individual determinant in the model was assessed from the significance level, the size of the unstandardized β-coefficient and standardized β-coefficient with the 95% confidence interval (95% CI), and the squared semi-partial correlations [33]. These squared semi-partial correlations indicate the unique contribution of each variable.

For all statistical analyses, the level of significance was set at *p* < 0.05 (two-tailed). Statistical analyses were performed using IBM SPSS Statistics for Windows, version 27.0 (IBM Corp, Armonk, NY, USA).

## 3. Results

### 3.1. Participant Characteristics

We combined data from 60 Belgian and 66 Danish community-dwelling persons at a median of 193 days (Q1–Q3 = 182–880) after stroke. Demographic and stroke-related characteristics of included persons are displayed in Table 2.

Data for sensor measures are presented in the lower part of Table 2. Included persons used their affected UL on average 3.88 h (SD 1.89). The activity ratio was 0.69 (SD = 0.27) and the variation ratio was 0.61 (SD = 0.29). The negative median magnitude ratio (median = −1.3; Q1–Q3 = −7 to −0.37) indicates increased activity of the unaffected UL relative to the paretic UL. The bilateral magnitude (median = 94.15 activity counts; Q1–Q3 = 68–114) indicates that the majority of UL activity was of low intensity.

### 3.2. Factors Associated with UL Activity

A standard multiple regression analysis was performed for each of the four dependent sensor measures with FMA-UE score, dominant hand affected, lesion side, type of stroke, age, gender, time post stroke, and country as independent variables. Assumptions were checked and resulted in a reflection and logarithmic transformation of FMA-UE and a logarithmic transformation of time post stroke. However, the results from the multiple regression with transformed variables did not differ from regression with non-transformed variables; thus, we decided to report the regression with non-transformed variables. No multicollinearity between independent variables existed. No cases had missing data, and no suppressor variables were found.

As seen from Table 3, FMA-UE score made a statistically unique contribution to all sensor measures. The high semi-partial correlations and large β-values further show that FMA-UE accounted for the largest percentage of the total variance of each sensor measure.

For model 1, FMA-UE accounted for 60.5% of the total variance for activity ratio, with an additional 3.3% explained by dominant hand affected. The adjusted *R*^2^ value of 0.663 (F(2,123) = 124.2, *p* < 0.001) indicates that more than two-thirds of the variability in activity ratio was accounted for by UL motor impairment and dominant hand affected. The size and direction of the relationship suggest that a higher activity ratio was achieved by those with higher FMA-UE scores and the dominant hand affected.

For model 2, along with FMA-UE (57.9%) and dominant side affected (3.6%), lesion side accounted for 1.8% of the total explained variance in magnitude ratio (66.8%, F(4,121) = 63.8, *p* < 0.001), with an additional 1.4% for type of stroke. The size and direction of relationships indicate that those with a higher FMA-UE score, the dominant hand affected, a right-hemispheric lesion, and ischemic stroke would reach a higher magnitude ratio.

The smallest amount of explained variance was found in model 3 with bilateral magnitude as a dependent variable (37.6%, F(4,121) = 19.81, *p* < 0.001). FMA-UE accounted for 18.1%, completed by age (6.2%), gender (3.5%), and type of stroke (2.9%). According to the size and the direction of the relationships, it is suggested that those who have a higher FMA-UE score, are younger and female, and had an ischemic stroke would achieve a higher bilateral magnitude.

Lastly, for model 4, the same independent variables as for activity ratio accounted for 58.7% of the variability in variation ratio, with 54% unique contribution by FMA-UE and 2.6% by dominant side affected (F(2,123) = 89.74, *p* < 0.001). As with the activity ratio, a higher variation ratio was achieved by those with higher FMA-UE scores and the dominant hand affected. The addition of time post stroke and country did not increase the total explained variance in any of the models.

## 4. Discussion

In this exploratory study, in a group of adults with chronic stroke, we developed models to determine variables associated with three aspects of daily UL activity represented by four sensor measures. Firstly, the contribution of each of the ULs to activity was represented by the activity ratio and the magnitude ratio. Secondly, the intensity of UL activity was represented by bilateral magnitude. Thirdly, the variation ratio gave insight into the variability of UL activity.

For all sensor measures, a large part of the variance (18–61%) was explained by FMA-UE, a measure of motor impairment [6,34]. This is in line with our hypothesis and with other work from the acute and chronic phase post stroke. In a group of 60 persons in the acute phase post stroke, sensor measures were compared between groups with different UL motor impairment [34]. Activity ratio, magnitude ratio, and variation ratio were lowest in the group with severe UL motor impairment, higher in the group with moderate impairment, and best in the mildly impaired group [34]. Similarly, in a group of 48 adults in the chronic phase post stroke, UL motor ability was moderately correlated with median magnitude ratio values [6].

A second important explanatory variable was dominant side affected. Although to a much smaller extent than FMA-UE, dominant side affected had a unique contribution to the explained variance of three out of four sensor measures. If the dominant UL was affected, persons showed a higher activity ratio and magnitude ratio (contribution to the UL activity) and a higher variation ratio (higher variability of UL activity). Accordingly, activity ratios and magnitude ratios were lower in persons in the chronic phase post stroke with the nondominant UL affected than in persons with the dominant UL affected [6,16]. Persons with their nondominant hand affected may be less motivated to use it as they still have a functionally intact dominant UL to complete daily activities.

FMA-UE score and dominant side affected accounted for the entire explained variance of activity ratio and variation ratio. The fact that activity ratio and variation ratio were explained by the same two variables to a similar extent is not surprising. Both sensor measures are ratios, calculated by dividing data on the affected UL activity by the unaffected UL (total hours of activity for activity ratio and acceleration variability for variation ratio).

For the magnitude ratio, along with FMA-UE and dominant side affected, an additional but minor part of the explained variance was accounted for by lesion side and type of stroke. Firstly, persons with a left-hemispheric lesion would have a lower magnitude ratio. This is somewhat contrary to what we would expect on the basis of the influence of lesion side on the paretic body side. As a left-hemispheric lesion results in a right hemiparesis and, thus, a dominant hand affected (87% of the cohort was right-hand dominant), we would expect a higher magnitude ratio. However, there might be other consequences of a left-hemispheric lesion resulting in a lower magnitude ratio such as limb apraxia, a common disorder associated with left-hemispheric stroke and a potent predictor of disability [35]. Secondly, persons with a hemorrhagic stroke would have a lower magnitude ratio. In contrast, previous literature showed a greater improvement in UL activity capacity in patients with a hemorrhagic stroke [36] and a better recovery of bimanual hand use [37]. As we did not consider lesion volume and exact lesion location, it might be that the lesion volume was higher and more often located in the UL (sub)cortical area in persons with a hemorrhagic, left-hemispheric lesion, resulting in lower UL activity outcomes.

For bilateral magnitude, the explained variance was much smaller despite more independent variables contributing to the model. Compared to the other three outcome measures, we were surprised to discover that FMA-UE explained much less of the total variance. Along with FMA-UE and type of stroke, age and gender were determining factors. A young woman would have a higher and, thus, better bilateral magnitude than an old man. This is in line with UL activity plots from the acute phase post stroke, indicating that the intensity of UL activity is generally higher in female patients with stroke compared to male patients [34]. Higher age was further associated with reduced levels of physical activity in adults with stroke [38] and might also have influenced the intensity of UL activity in our sample. Contrary to the models for the other three sensor measures, dominant hand affected was not retained as a significant independent variable for bilateral magnitude. Similarly, in another cohort in the chronic phase, the median bilateral magnitude was not different between patients with the dominant side affected versus patients with the nondominant side affected [6]. The median bilateral magnitude was further only weakly correlated with motor capability [6]. Accordingly, in our study, affected UL motor impairment explained only 18% of the explained variance in bilateral magnitude, which is lower than the 54–61% for the other three sensor measures. It is likely that the size of bilateral magnitude, calculated as the summed intensity of both ULs, is mostly influenced by the unaffected UL.

As the included independent variables accounted for not more than one- to two-thirds of the variance in UL activity, other factors may contribute. Firstly, time spent in sedentary activity was associated with hours of UL activity in nondisabled adults [5] and might also be associated with the sensor measures in our sample. A second factor that might have been relevant is ADL dependence, as higher dependence in ADL was associated with lower activity ratios in a chronic stroke sample [16]. Thirdly, sensory function may also play a role as it was previously shown to be a significant predictor for bimanual hand use in the chronic phase post stroke [37]. Lastly, in a small group of 20 patients with chronic stroke, lower attention and arousal predicted low affected UL activity above and beyond sensorimotor impairment [39]. Behavioral factors such as apraxia, neglect, attention, and self-efficacy could also have influenced our sensor measures and should be investigated further in larger sample studies.

Some limitations should be mentioned. Firstly, we recognize the fact that accelerometers do not exclusively reflect functional movements and that there are some barriers to the use of available wearable sensor technology [4]. However, for the time being, accelerometry seems to be the best available means to measure UL activity in daily life, outside the structured environment of the lab. Secondly, the cross-sectional nature of the study implies that we could not demonstrate a cause–effect relationship between associated factors and the different aspects of UL activity. Our results can only be generalized to community-dwelling individuals in the chronic stage of stroke recovery. Lastly, pooling data between a Belgian and Danish cohort led to a loss in accelerometry data (12 h interval extracted from 72 h accelerometry data from Belgian cohort) and potential discriminating factors (only those kept that were available in both cohorts). However, pooling is also an important strength of the study, as it resulted in a large sample size (*n* = 126). This large sample size allowed us to perform multiple regression, which included several independent variables. Furthermore, the large sample comprised persons over the entire range of UL motor impairment (FMA-UE min 0 max 66), representative for the stroke population, with a similar distribution of gender, dominant side affected, and lateralization.

As the goal of rehabilitation is to improve UL activity in daily life, clinicians need to quantify UL activity. Wrist-worn accelerometry is a reliable and valid way to quantify UL activity in adults with stroke [11,13,14] and can provide feedback on a patient’s UL activity during daily life. Despite the evidence for the use of extrinsic feedback in recovery [40] and the use of accelerometry to quantify UL activity, wrist-worn accelerometry is not widely used in routine clinical practice [4]. One of the barriers to its adoption in clinical practice is that many sensor measures capture similar or related aspects of activity, and it is not known which measure is best used [4]. In this study, we highlighted that the measure of choice depends on the aspect of UL activity of interest. If a general overview of the duration of UL activity is wanted, activity ratio is a good choice, as it has strong psychometric properties [5,11], has established norm values [9,12], and is easy to understand. If, on the other hand, we want a better view of how both limbs are used together in terms of contribution to and intensity of UL activity, magnitude ratio and bilateral magnitude may add value. Although these measures are less straightforward and a bit more challenging to understand, a visual overview of both measures can be given in a density plot [29]. Secondly, the choice of a sensor measure may depend on what the measure is used for. Whereas bilateral wrist-worn accelerometry reflects the quantity of UL activity, full-body motion capture suits (including more sensors on several body segments) might give better information on the quality of UL activity [41]. However, full-body motion capture suits are less suitable for long-term measurements than bilateral wrist sensors. Therefore, sensor measures from wrist-worn accelerometry that do not only reflect quantity but also the quality of UL activity might be interesting to distinguish compensation versus restoration [15]. It was recently shown that persons who move the ULs more often and with more variability tend to move with fewer compensations [15]. Activity and variation ratio appear to reflect not only quantity but also quality. However, we showed that activity ratio and variation ratio were explained to the same extent by the same variables. Furthermore, as variation ratio is less well studied than activity ratio [4], information about variation ratio might be less straightforward to use in clinical practice. Lastly, as the percentage variance explained by UL motor impairment differed between sensor measures, it might be interesting for future research to see how improvements in UL impairment translate into changes in the different aspects of UL activity. As UL impairment only explained a part of the variance in the different sensor measures, future research should consider investigating the influence of behavioral strategies in rehabilitation, which may address different aspects of UL activity. These efforts might include behavioral methods, such as contracting, self-monitoring, problem-solving, and home skill assignments [42,43]. Furthermore, it may be important to personalize rehabilitation and regularly give feedback, which is associated with better activities in daily living and reduced nonuse [39,44,45].

## 5. Conclusions

The present study contributes to a better understanding of what influences different aspects of UL activity in the chronic phase post stroke, which may further guide strategies to improve UL activity and, thus, enhance the performance of everyday tasks. As hypothesized, all aspects of UL activity were mainly explained by UL impairment. However, the amount of variance explained by UL impairment, as well as the total amount of explained variance, depended on the specific aspect of UL activity and was lowest for the intensity of UL activity. We conclude that, when choosing an appropriate sensor measure, it is important to consider the specific aspect of UL activity one is interested in. Furthermore, a substantial part remains unexplained by UL impairment and biological factors, which warrants further investigation. As precision medicine is getting more attention in today’s research and clinical practice, personalized rehabilitation including behavioral strategies might be important to find the best fit for each individual.

## Figures and Tables

**Table 2 sensors-22-02273-t002:** Demographic and stroke-related characteristics presented as medium (SD), median (Q1–Q3), or number (%).

	Belgian Cohort (*n* = 60)	Danish Cohort (*n* = 66)	All Participants (*n* = 126)
Age at inclusion (years)	59 (13)	66 (10)	62 (12)
Gender (male), *n* (%)	37 (62)	40 (61)	77 (61)
Days since stroke onset	977 (577–1618)	182 (179–185)	193 (182–880)
Stroke etiology (ischemia), *n* (%)	37 (62)	54 (82)	91 (72)
Lateralization (left hemisphere), *n* (%)	28 (47)	31 (47)	59 (47)
Dominant side affected, *n* (%)	27 (45)	33 (50)	60 (48)
Hand dominance (right), *n* (%)	52 (87)	58 (88)	110 (87)
Living arrangement, *n* (%)			
Living alone	29 (48)	48 (73)	77 (61)
Living not alone	31 (52)	18 (27)	49 (39)
FMA-UE	53 (27–62)	58 (47–63)	57 (34–62)
Affected UL use	3.80 (1.88)	3.95 (1.91)	3.88 (1.89)
Activity ratio	0.66 (0.27)	0.72 (0.27)	0.69 (0.27)
Median magnitude ratio	−2.06 (−7–−0.51)	−0.93 (−3.11–−0.35)	−1.30 (−7–−0.37)
Median bilateral magnitude	90.99 (66.24–114.72)	95.39 (73.11–114.01)	94.15 (67.90–114.01)
Variation ratio	0.57 (0.29)	0.64 (0.28)	0.61 (0.29)

FMA-UE: Fugl–Meyer Assessment upper extremity.

**Table 3 sensors-22-02273-t003:** Standard multiple regression models to examine the association between various independent factors and different sensor measures.

		Unstandardized Coefficients		Standardized Coefficient			Change Statistics			
Dependent Variables	Independent Variables	B	SE	β	95% CI for β	Significance	Adjusted Model *R*^2^	SEE	*sr*^2^ (Unique)	Significant F Change
Activity ratio	FMA-UE	0.011	0.001	0.781	0.685 to 0.872	<0.001 *			0.605	
	Dominant side affected	0.098	0.028	0.181	0.081 to 0.280	<0.001 *			0.033	
	Constant	0.140	0.038			<0.001 *				
	Model						0.663	0.157		<0.001 ꭛
Magnitude ratio	FMA-UE	0.108	0.007	0.769	0.665 to 0.869	<0.001 *			0.579	
	Dominant side affected	1.534	0.418	0.277	0.106 to 0.462	<0.001 *			0.036	
	Right-hemispheric lesion	1.083	0.418	0.195	0.027 to 0.383	0.011 *			0.018	
	Hemorrhagic stroke	−0.742	0.321	−0.120	−0.230 to −0.017	0.023 *			0.014	
	Constant	−8.780	0.551			<0.001 *				
	Model						0.668	1.601		<0.001 ꭛
Bilateral magnitude	FMA-UE	0.673	0.111	0.432	0.290 to 0.574	<0.001 *			0.181	
	Age	−0.642	0.182	−0.254	−0.397 to −0.111	<0.001 *			0.062	
	Male	−12.001	4.586	−0.191	−0.335 to −0.046	0.010 *			0.035	
	Hemorrhagic stroke	−11.745	4.904	−0.172	−0.341 to −0.030	0.018 *			0.029	
	Constant	113.330	13.061			<0.001 *				
	Model						0.376	24.309		<0.001 ꭛
Variation ratio	FMA-UE	0.011	0.001	0.738	0.654 to 0.836	<0.001 *			0.540	
	Dominant side affected	0.094	0.033	0.163	0.055 to 0.273	0.006 *			0.026	
	Constant	0.055	0.045			0.222				
	Model						0.587	0.185		<0.001 ꭛

95% CI: 95% confidence interval; SEE: standard error of estimate; *sr*^2^: semi-partial correlation. * The β-coefficient was statistically significant. ꭛ The model *R*^2^ change was statistically significant.

## Data Availability

The data analyzed during the current study are available from the corresponding author, Bea Essers, upon reasonable request.

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
