# Peer review of "Determinants of Different Aspects of Upper-Limb Activity after Stroke"

_sensors, 2022, doi:10.3390/s22062273_

Round 1

Reviewer 1 Report

Thank you for opportunity to review this interesting study.

  1. You mention the data loss from pooling the measurements (lines 348-354) as a potential limitation. Though individually the pools may be under-powered, was any analysis done to look at the pools individually (i.e. would your conclusions have been the same looking at each pool separately)?
  2. The main point of the discussion (lines 381-384) seems to be that behavioral factors may by a large influence in the variance of the sensor measurements and that future studies should focus on this. Can you provide any examples as to what kind of strategies this may entail?

We congratulate the authors on their study and its potential important impact on future research in data-driven rehabilitation methods.

Reviewer 2 Report

Thank you for the opportunity to review the manuscript: Determinants of Different Aspects of Upper Limb Activity After Stroke, it was a exploratory study was to establish associative factors of different aspects of UL activity in a larger group of adults with chronic stroke. The hypothesiz that UL impairment will be a significant determinant of all aspects of daily UL activity and that other factors will contribute, such as gender and whether the dominant hand is affected. Improved understanding of what influences different aspects of UL activity in the chronic phase post-stroke might further guide strategies to improve UL activity and thereby enhance the performance of everyday tasks in the chronic phase post-stroke.This manuscript is very interesting and makes a valuable contribution to research into post-stroke functional monitoring in home rehabilitation settings. Nevertheless, I have a few comments:
1. Were the participants tested for cognitive impairment?
2. I suggest, characterize the ADL activities that were performed by the participants, because their type was crucial in the presented measurements (excluding FMA-UE activities)
3. If possible, please add the following work to the discussion:
Held JP, Klaassen B, Eenhoorn A, Beijnum B-JFv, Buurke JH, Veltink PH, Luft AR. 2018. Inertial sensor measurements of upper-limb kinematics in stroke patients in clinic and home environment. Frontiers in Bioengineering and Biotechnology 6:27
4. The conclusions are very general, please the authors to refer to the obtained results and the purpose of their research.
